# Improved Multimedia Object Processing for the Internet of Vehicles

**DOI:** 10.3390/s22114133

**Published:** 2022-05-29

**Authors:** Surbhi Bhatia, Razan Ibrahim Alsuwailam, Deepsubhra Guha Roy, Arwa Mashat

**Affiliations:** 1Department of Information Systems, College of Computer Sciences and Information Technology, King Faisal University, Al-Ahsa 31982, Saudi Arabia; raalsuwailam@kfu.edu.sa; 2Department of Computational and Data Sciences, Indian Institute of Science, Bangalore 560012, India; roysubhraguha@gmail.com; 3Faculty of Computing and Information Technology, King Abdulaziz University, Rabigh 21911, Saudi Arabia; aasmashat@kau.edu.sa

**Keywords:** edge intelligence, multimedia object processing, cooperative learning, industrial Internet of Things

## Abstract

The combination of edge computing and deep learning helps make intelligent edge devices that can make several conditional decisions using comparatively secured and fast machine learning algorithms. An automated car that acts as the data-source node of an intelligent Internet of vehicles or IoV system is one of these examples. Our motivation is to obtain more accurate and rapid object detection using the intelligent cameras of a smart car. The competent supervision camera of the smart automobile model utilizes multimedia data for real-time automation in real-time threat detection. The corresponding comprehensive network combines cooperative multimedia data processing, Internet of Things (IoT) fact handling, validation, computation, precise detection, and decision making. These actions confront real-time delays during data offloading to the cloud and synchronizing with the other nodes. The proposed model follows a cooperative machine learning technique, distributes the computational load by slicing real-time object data among analogous intelligent Internet of Things nodes, and parallel vision processing between connective edge clusters. As a result, the system increases the computational rate and improves accuracy through responsible resource utilization and active–passive learning. We achieved low latency and higher accuracy for object identification through real-time multimedia data objectification.

## 1. Introduction

Human-like learning is one of the key challenges in the area of computer vision [1]. The response rate of and bypassing an accident for an adult are faster than a child, because an adult is more experienced than a child. However, children could act better than adults if they are trained for a certain period of time in particular circumstances. Similarly, a machine can be more accurate through simultaneous case testing and learning procedure. This procedure can even be more rapid if we make an intelligent edge cluster among the present devices in the same local network. This work aims to connect all the incorporated sensor end-devices of a smart vehicle, collecting several real-time multimedia data. The data are processed further into the cluster using active and passive learning. In the last decade, the assortment of edge computing and deep learning has helped to make edge devices intelligent. Intelligent edge devices can make several conditional decisions using machine learning algorithms that are comparatively secure and fast. In addition, data analysis has been widely adopted and is becoming the center of big data creation in every industrial IoT sector, such as automobile, robotics, management, and automation [2,3,4,5]. In most cases, the automation industry is remodeling predefined or existing frameworks, instead of building a computerization model from scratch [6]. Similarly, our motivation is to design an efficient multimedia data collection and dispensation model to obtain accurate and faster multimedia data processing for a smart vehicle. We selected an advanced boundary box iteration model [5] and cooperative deep learning to integrate edge nodes with similar object-detection capacities. The semantic interoperability among the intelligent nodes makes the overall procedure time efficient.

In the case of a traditional surveillance camera in a vehicular model, the collected data is stored on a local or remote cloud server for further utilization [4]. On the other hand, the smart surveillance camera model utilizes the collected video data for real-time automation, such as threat detection in industries. We decided to distribute the computational load by slicing our object data among analogous intelligent nodes. The model produces a parallel multimedia processing platform among the connective edge clusters. After completing the object-detection method, the sliced tasks are stitched on the cluster head. We used the prediction model based on the boundary box creation for multimedia data processing and compared the accuracy and propagation time with existing solutions. As a result, we reduced the computational time by distributing the tasks and transferring knowledge among interconnected edge devices. In the object detection phase, we predict the next position of an individual object considering its prior status. The model has been trained to process audio data processing parallel to image detection. The system increases the computational rate and improves the accuracy through trustworthy resource utilization. Each smart vehicle acts as a cluster of receiver ends in this procedure. The receiver ends are capable of performing the fundamental measures toward machine learning and accomplishing the following contributions:Computation and validation of received real-time data.Cooperative processing of multimedia data.IoT-facts handling, such as low light or high-frequency object finding.Accurate detection.Real-time decision making, such as avoiding accidents or not taking over an ambulance.

Therefore, the overall contributions of our proposed model in terms of an intelligent network in industrial IoT are as follows:The corresponding comprehensive network combines cooperative multimedia data processing and reduces the tension to publish/offload the data on the public cloud for further processing.The ubiquitous computing structure inspires the proposed model. After reaching a certain threshold, the computational load is distributed by slicing real-time object data among analogous intelligent IoT nodes, and parallel vision processing between connective edge clusters occurs.Cooperative machine learning helps to reduce compilation errors and increases trust among the decision-making units.IoT fact-handling faces resource thrusting during manual or automatic triggers. Edge-intelligent-based cooperative learning can significantly reduce it.The model increases the computational speed and validates the real-time object by near-optimal 3D processing.The backbone network is ubiquitous-computing based, making the IoV connectivity more accurate to human-like decision making.IoV network faces throughput deficiencies, while offloading data to third-party servers. These actions confront real-time delays during data offloading to the cloud and synchronizing with the other nodes. The proposed model helps to increase the throughput efficiency.An intelligent edge cluster does not claim to block data sharing or offloading to the distance server.

As a result, the system increases the computational rate and improves the accuracy by responsible resource utilization and active–passive learning. At the same time, possibilities of data breaching, data clashing, and data stealing are reduced, compared to cloud-based offloading. Figure 1 shows the overall task flow into the proposed cooperative learning model for task slicing and load distribution among similar capacities having edge devices.

The organization of the paper is as follows: Section 2 is a comprehensive discussion of the related works, including distributed offloading, intelligent edge cluster, decision-making module, multimedia data processing, and service experience matrices; Section 3 highlights the proposed advanced edge computing algorithms; Section 4 is the result analysis section; and Section 5 concludes the paper.

## 2. Related Work

Multimedia data computation and object detection at the edge is a concept that has already been explored [4]. A few studies have already explored how to bring an intelligence camera to compute real-time data processing [5,6,7]. Some have deployed hardware-accelerated custom application-specific integrated circuits (ASICs) and field-programmable gate arrays (FPGAs) to accelerate the overall processing. They were also performed with a low-power configuration strategy by bio-inspired event-based processing. Whereas these standalone intelligent cameras are highly acceptable for specific functions, such as person detection or classified object detection using vision processing, our motivation is to design an improved algorithm to facilitate a smart car into an IoV network. Existing object detection and machine learning approaches exist in a statically defined local network and are connected with a private/cloud server. In contrast, we propose a load-distribution and decision-making strategy, while a smart car continuously moves from one place to another. Standalone intelligent camera modules installed in industrial areas are possibly connected using a similar protocol or application gateway. Additionally, the computational task was performed by the edge device itself.

However, computational load on a single module is not time efficient in many real-time task offloading, for example, for a moving car. Therefore, efficient task diffusion could be helpful in service quality enhancement in terms of time and accuracy. On the other hand, the boundary box optimization method has gained popularity in real-time object detection [8,9]. This particular methodology has been proved to be appropriate for a lightweight edge module. Nevertheless, there is a time mismatch for dynamic object detection if we use a single-edge node that acts as a deep learning module. We can increase the quality of service by distributing the load among similar computational capabilities having nodes. Therefore, we propose a slicing and stitching methodology using an advanced bounded box iteration model to interact, allocate, and accommodate heavy computational tasks, such as real-time multimedia data processing. The novelty of our model is to reduce compilation errors and increase trust among the decision-making units. This section introduces a few related technologies and strategies that crucially assist our proposed approach.

### 2.1. Distributed Offloading

In the traditional offloading method, the task is offloaded to the best cloud for further execution, and the consequent value comes back to the resource after the completion of the performance. In comparison, the cloud has far more processing units and capacity than edge servers [3,9]. Therefore, distributed offloading into the edge cluster needs more than one surrogate server. Henceforth, unlike one-to-one task offloading in a cloud-based approach, the edge-cluster approach follows distributed or one-to-many offloading strategies. Distributed offloading needs more complicated load distribution strategies than cloud-based offloading; however, in a network of intelligent things, the end nodes come with several constraints, such as a short battery life, low computational capacity, and lightweight data security. Therefore, edge cluster-based computing is more suitable for low-bandwidth-supported IoT networks.

The low latency, quick response, and accurate processing of real-time multimedia data are the primary entities of an automated car. As sensors, cameras and audio adopters are the sources of real-time elements; the automated vehicle can make an edge cluster for a predominantly secured processing unit itself [4,5]. Distributed offloading to the cluster head for further load distribution is one of the best solutions for IoVs. Distributed learning and processing can provide intelligent edge clusters with the perfect consummation of quality-of-experience matrices.

### 2.2. Intelligent Edge Cluster

Intelligent edge is the most suitable option for resource starving IoV networks. Every cluster has one cluster head connected to the other cluster heads and IoT nodes in the intelligent edge cluster network [8]. Eventually, they are the decision makers and load distributors of each collection [9]. A fundamental task, such as image processing, audio processing, and sensor data processing (for example, humidity, temperature, and light) can be performed individually by the edge-IoT device itself. However, when it comes to an automated vehicular system, a cluster could be made of all the multi-processing units responsible for a particular vehicle.

Similarly, other cars have their intelligent edge cluster for making decisions in real time. One intelligent micro-edge of each group is nominated as a cluster head, which is fixed and manually determined during installation. Each cluster designed for each car is smart enough to make a decision in real time with parallel active–passive learning. The processing and decision making concerning the edge make an IoV unit more trustworthy by itself. Possibilities of data breaching, data clashing, and data stealing are reduced, compared to cloud-based offloading. An intelligent edge cluster does not claim to block data sharing or offloading to the distance server [10]. Nonetheless, it helps to decide to share information in a reliable way with the other resources. Similar to the case of petrol-pump finding, the car has to share its current location. Here, we come to the conception of active–passive learning and cooperative learning.

### 2.3. Active–Passive Learning and Cooperative Learning

An intelligent cluster can perform two things simultaneously. Active learning is making decisions within the cluster and passive learning occurs during the knowledge sharing among the other cluster heads. Passive learning can occur in the subconscious core with little web consumption to decrease bandwidth constraints [10]. An automated vehicle needs to gather multimedia data from different edge-IoT servers, which can make a diminutive decision, such as adjusting the brightness of headlights and not sounding the horn while passing a hospital or comprehending the traffic lights and signals. The concept of clustering provides a semantic-based learning environment to gather and distribute knowledge among the edge clusters. This knowledge sharing is called cooperative learning. Another major aspect of passive learning is establishing trust among the IoV nodes so that, in the instant of an accident, the decision maker can ask for/receive help from the trustworthy node. This reliable node could be a smart ambulance, smart loader, or even a smart police van. Active learning is a more accurate, problem-solving, and self-involved procedure depending on the installed applications, algorithms, and decision-making capacity. Furthermore, the accuracy of any established algorithm can be increased by continuous and cooperative learning and active–passive data processing.

### 2.4. Multimedia Data Processing

The proposed model follows many-to-one mapping to collect multimedia data from different intelligent edge nodes [11,12], such as audio data from smart audio-receiving nodes, image data from differently placed smart image-processing nodes, and real-time sensor data from smart sensor nodes, assembled to the cluster head. The cluster head distributes the collected data to the connected, intelligent edges using the slicing mechanism. Here, every participant-edge server is equally prioritized and has an equivalent computation capacity.

Multimedia data processing includes audio–visual data with the addition of corresponding sensor data. Multimedia data processing has basic targets depending upon shallow or individual decision-based node learning and deep learning-based cooperative decision making by more than one node or edge cluster [13]. As previously explained, a smart car can decide not to sound a horn while hospitals or schools are nearby, only utilizing the location-based application. We can optimize this decision by using more than one source data; it effectively includes audio- or image-based data. If the school day has ended or it is the school break time, and the traffic is also significant, intelligent edge clusters can optimize the actions. The smart car can find the person we are looking for without creating any disturbance, or it can effectively reduce the sound of the horn. It can also share the real-time update of traffic to its trustworthy cluster heads.

### 2.5. Decision-Making Module

Decision-making modules are essential to reduce the computational load, optimize the service matrix, and the overall network trust. Here, the term decision-making module has been used alternatively for a single intelligent edge and cluster head as per their operational perspective.

Two kinds of task handling have been conducted in the proposed cooperative model. While there is no manual or automated load triggering, the system continuously performs passive learning; after a manual or automatic load triggers the IoV module, it shifts to the active-learning procedure. The node itself acts as a decision-making module for a singular matrix handle related to image processing, audio processing, or sensor data handling by individual intelligent edges [8,14]. For passive-learning conditions (mentioned in Algorithms) cluster heads act as decision-making units to decide which information should be shared within the cluster and which data could be published among the other cluster heads of a more extensive IoV network. The cluster head also distributes the load by choosing the appropriate data handler and allocating the computational task.

### 2.6. Service Experience Matrices

QoE or quality of experience matrices are defined to match the service matrices determined by a particular computational model or service provider. An intelligent IoV network is similarly intended to uplift sustainability and reduce existing shortcomings similar to other industrial IoT model. The quality-of-service pyramid consists of service cost, reliability, latency, throughput, security, and other network-related constraints [15,16]. In comparison, the QoE aspect is defined to fulfill the desired QoS and reach the top of the QoS pyramid. The evaluation of Industry 4.0 already recognizes user perspective and satisfaction, which highly demand sustainable QoE. If a model continuously fails to match the user demands, other solutions can easily replace it, as the market has several innovative resolutions.

We decided to reduce the computational limitation by slicing real-time multimedia data among analogous intelligent edges. The model obtained a parallel multimedia processing medium among the connective edge clusters. After completing the object-detection method, the cluster head compared the sliced tasks and optimized the decision. We used the prediction model based on the edging box composition for multimedia data processing and compared the accuracy and propagation time with existing solutions. Consequently, we reduced the computational time by distributing the tasks and transferring knowledge among interconnected edge devices. In the object-detection phase, we predicted the next position of an individual object considering its prior status. The system increased the computational rate and improved its accuracy by responsible resource utilization. Table 1 summarizes some of the existing literature, along with their contributions and areas of application.

## 3. Advanced Edge Computing Algorithms

Edge computation is more reasonable than the conventional VM-ware-based cloud computing to connect the resource-thrusting IoT nodes. The primary conception of making an edge cluster is similar to the software-defined cloud cluster; it helps to avail the high computational intelligent-capacity destination near the edge.

Analysis, innovation, and AI-based automation have helped grow smart edge clus-ters to serve fast, reliable computational platforms in the last decade. Here, the entire network can perform active-learning and decision-making steps from Algorithm 1, while fast multimedia data processing. An individual node can make a short-term decision depending upon self-analysis capacity by following Algorithm 2, step 6 onwards. The edge cluster can also perform cooperative learning among the cluster heads using the passive-learning mode, explained in Algorithm 3. The decision making unit decide and entire procedure which could be the near optimized way for that particular situation handling.
**Algorithm 1:** DMO algorithm**• Input:***n* numbers of data input matrices from *n* number of different edge devices**• Output:** Decision-making optimization1: Start screening by the cluster head2: If Passive learning is running{  Check speed of the vehicle and area congestion   {    If (i.      Current_speed => Threshold_speed &&ii.     Current_congestion=>Threshold_congestion ){ 3: Initialize the Active learning;4: Increase the processing capacity;5: Check status of the receiver ends;6: Check the nature of the work load;    }    If (forceful and manually urgent for particular     activity)    {     Call Algorithm 2 for singular matrices data       processing;    }    Else: Call Algorithm 2 for multimedia matrices         data processing;   Else: Continue passive learning; }7: Check the status of the receiver nodes regularly;8: Initialize parallel processing into the edge cluster;9: Train the DNNs and update;11: End;}

**Algorithm 2:** NOLD algorithm**• Input:***n* numbers of data input matrices from *n* number of different edge devices**• Output:** Near optimal load distribution1: Follow Algorithm 1 up to step 2;2: Initialization of Active learning:3: Prioritize each edge;4: Replicate different workloads to all edges;5: Call Algorithm 2 for singular matrices data  processing:6: Check the nature of the work load;For singular matrices data processing: If (Data=image_data){    Distribute the load && apply boundary box     object detection;  }  If (Data=audio_data){    Distribute the load && apply audio-based     object detection; }  If (Data=sensor_data){    Distribute the load && apply condition-based     operation handling; }  Else: Call Algorithm 2 for multimedia matrices  data processing;End forFor multimedia data processing:7: Increase the processing capacity of the GPUs;8: Start thread handling to optimize process utilization;9: Check the status of the request handler regularly;  If (i.     Current_speed < Thresold_speed &&ii.    Current_congesion < Thresold_congesion){ 10: Initialize the Passive learning;11: Train the DNNs and update;12: End;  }

Algorithms 1 and 2 help to perform the decision-making procedure. At the start of the connection of the IoV node to the IoV network, the already decided cluster node starts the screening procedure. By default, the passive learning starts, which is described by Algorithm 3. As per Algorithm 1, the intelligent edges continuously sense the speed of the vehicle and congestion or traffic volume of the external world. Here, one self-efficient smart vehicle is considered as the IoV node. The similar node is also considered as an edge cluster. If the current speed of the IoV module is more than the threshold speed and the current congestion is more than its threshold value, then the active learning is initiated by the cluster head. It helps to proceed further, according to Algorithm 1.
**Algorithm 3:** Passive-learning algorithm**• Input:***p* number of different cluster heads**• Output:** Trust build and optimization1: Start learning:2: Manually set cluster head for individual edge cluster;3: Assign individual tasks while designing the IoV network;4: For trust building: { If (Current_load < threshold_load && no manually   triggered service request){    Share globally requested data to build the trust; }  If (Current_load < threshold_load && manually    triggered service request){i.     Act as per the request;ii.    Search for incorporated solution;iii.   Utilize previously built trusts into the IoV network;   }  If (Current_load > threshold_load && automatically    triggered service request){    Distribute the load && apply condition-based     operation handling; }  Else: Call Algorithm 1 for multimedia matrices  data processing;End for9: Train the DNNs and update;10: End;  }

Figure 2 depicts the system flow while producing certain conditional approaches for the load distribution. Primarily, an individual edge cluster, which is also a self-sufficient smart vehicular system, enters into the largely connected IoV system. Subsequently, the possible task flows can occur. The entire direction can be divided into six levels, as shown in Figure 2. Explaining the diagram using the top-to-bottom approach, level 1 encounters the initialization phase of the entire procedure. The moment an individual IoV node enters the IoV network, three different possibilities arise. The first one is to select Algorithm 1. The next one is to choose the unknown idle period if the connectivity cannot be correctly established or the user wants to exit the network. The third possibility is to handle the smart vehicle manually, without the absence of sufficient network resources, or intentionally.

At the next level, there are two possible occurrences: one follows step 6 from Algorithm 2, to make individual decisions for singular matrices data, such as audio; the other is to select cluster-head activation as the decision-making module for further procedures.

At level 3, the task module can make two different decisions: one is to decide upon the active-learning procedure as per Algorithm 1, from step 3 onwards; the other is to select the near-optimized computation of singular matrices data by assigning the load to similar kinds of computational edge units.

At level 4, the active learning and task distribution processing starts as per Algorithm 1. The procedure makes sure to select load distribution as per the system’s requirements. At this level, there is an extra provision to choose singular matrices data processing in addition to the near-optimal multimedia data processing.

At level 5, first, stitching procedures occur. This is for decision optimization after the processing of singular matrices. Other flows occur for multimedia data processing and optimization.

At level 6, the other stitching procedure occurs. This is for decision optimization after the 3D-data processing for multi-matrices.

Finally, the near-optimized decision making starts and continues until the IoV node is disconnected from the IoV network either manually or accidentally.

Figure 3 depicts the particular work-flow diagram for multimedia data processing and handling using Algorithms 1 and 2. It shows how the cluster head (CH) acts as the decision-making module and distributes the triggered load among similar kinds of data-processing units. From Figure 3, it can be included that the decision-making module works in a self loop to incorporate updated information, while making a decision on load balancing.

## 4. Result Analysis

We used the prediction model based on boundary box creation for multimedia data processing and compared the accuracy and propagation time with the existing solutions. As a result, we reduced the computational time by distributing the tasks and transferring knowledge among the interconnected edge devices.

For the purpose of the result analysis, we took a real-time video-processing challenge, where we considered three problem-defined scenarios. We followed the work-flow diagram of the proposed edge intelligence-based cooperative learning model with a multimedia object-processing handler in the industrial Internet of Things shown in Figure 3. Table 2 presents an idea of the components of our prototype model.

We observed the before-task allocation status of the overall edge cluster. Then, we manually triggered the cluster head for the rapid processing of the multimedia data. Then, we considered the output data for single matrices data, which were image data. Another factor we considered was the audio–visual multimedia data for video processing. In Figure 4, we can observe the status of the processing unit just before the start of the cooperative-based learning procedure (at Level 3, Figure 4, top-to-bottom approach). The left side of Figure 4 clearly shows that no processing has been started. The processing unit utilization was 16% at this point, which means that the cluster head had already started its function as a decision-making unit.

Figure 5 depicts the process-utilization phase when we follow Algorithm 2, step 6 onwards for the image processing procedure. The audio processing step also runs parallel to the other edge node. Furthermore, the sensor nodes continuously process the environmental data, such as the temperature and humidity. We specifically focused on the multimedia task scheduling, processing, and utilization of the service module, while following our proposed slice and stitch model.

Figure 6 depicts the overall/shared processing effort towards audio–visual data processing. The utilization increased from 16% to 28%, while processing the video data. Shared memory utilization also increased by 3.08%.

We entered seven intelligent edges into our prototype IoV model for the single IoV node. Then, we assigned one edge as the cluster head and connected the other six edges to that cluster head. Figure 7a shows the overall processor utilization for video-data processing by the cooperative-based learning approach. Figure 7b shows the service processor utilization of mother edge or the data-receiving end, whereas Figure 7c–h show the utilization of edge devices for video-data processing by the cooperative-based learning approach from edges 1 to 6. Figure 8a shows the utilization of cluster-head memory, while making decisions for near-optimized video-data processing by the cooperative-based learning approach. Figure 8b shows the utilization of virtual memory at the cluster head, while making decisions for near-optimized video data processing by the cooperative-based learning approach. Figure 8c depicts the hard faults/sec measurement by the cluster head, while distributing the load into the edge cluster for near-optimized video data processing by the cooperative-based learning approach.

Figure 9a shows the data-allocation rate from receiver end to the processing edge while distributing the task by the cluster head for near-optimized video-data processing by the proposed cooperative-based learning approach. During the distribution, the active time was 38% of the total processing time. The read speed was 472 kb/sec from the mother edge, while the write speed to the surrogate edge was 91.0 kb/sec. The average response time after the overall processing was 11.6 ms.

Figure 9b shows the throughput of the used network at a 30 s duration. This throughput was fixed for all the edges, as the edge cluster shares the same Internet availability.

Figure 10a,b show disk utilization for a similar, but reverse, condition during the data allocation from the surrogate end to the mother edge in the presence of the cluster head for near-optimized decision making by the proposed cooperative-based learning approach. Figure 10b shows the repetitive collection of processed data from edges 1 to 6 at a 60 s duration. This procedure followed the cluster-mapping procedure, which is also called the data-allocation tree.

We conducted a comparative analysis of training loss/accuracy and validation loss/accuracy of an existing deep learning model. Figure 11a,b show the resultant curves for the same model.

The receiver operating characteristic graph presented in Figure 12 depicts the implementation of a classification model for all classification thresholds. This graph shows the different types of variations with their corresponding values. This plot generally reflects two values those are the true positive rate and true negative rate value. We used the prediction model based on boundary box creation for multimedia data processing and compared the accuracy and propagation time with the existing solutions. As a result, we reduced the computational time [17] by distributing the tasks and transferring knowledge among the interconnected edge devices. In the object-detection phase, we predicted the next position of an individual object considering its prior status. The model was trained to process audio-data processing parallel to image detection. The system increased the computational rate [18,19] and improved the accuracy rate by responsible resource utilization.

Table 3 shows a comparative analysis of an existing model with our proposed prototype.

## 5. Conclusions

Industry 4.0 has shown how an appropriate combination of edge computing and deep learning makes edge devices intelligent [20,21]. Intelligent edge devices can endure several conditional determinations using machine learning algorithms that are comparatively secure and fast. In addition, data analysis has also been highly recognized for analyzing large amounts of data in every industrial IoT sector, such as automobile, robotics, management, and automation [22,23,24,25]. We remodeled the existing Internet of vehicle system using comparatively small, but powerful, edge clusters. It can increase the conquest rate and reduce the building cost of a computerization model from scratch.

We collected real-time data to obtain more accurate and faster multimedia data processing in intelligent vehicles using a multi-dimensional data collection and dispensation model. We chose to reduce the computational limitation by slicing real-time multimedia data among analogous intelligent edges. The model used active–passive learning to obtain a parallel multimedia processing medium among connective edge clusters. After completing the object-detection method, the cluster head compared the sliced tasks and optimized the decision. We obtained the following conquest:The success rate of the prototype model was satisfactory, with several position changes and speeds.Load distribution was also adequate while increasing the congestion.Cluster-head selection was dynamic, but cost-efficient in terms of power consumption.

We targeted a few additions and plan to improve our prototype in the following ways:The prototype model needs to be integrated as a user-friendly product.We primarily used the ‘secure boot’ technique to secure our edge devices. We plan to use blockchain security for connected edge clusters in the future [26].We plan to integrate vehicles in the food-supply chain with the concept of a sustainable supplier selection based on the concept of Industry 4.0 [27].We plan to increase scalability by incorporating novel metaheuristics algorithms, such as the red deer algorithm [28], whale optimization algorithm [29], and social engineering optimizer [30], in the future.We plan to use semantic interoperability to create intelligent ambulatory systems for the smart healthcare industry [31].We plan to add more sensor devices, while keeping the budget market reasonable.

## Figures and Tables

**Figure 1 sensors-22-04133-f001:**
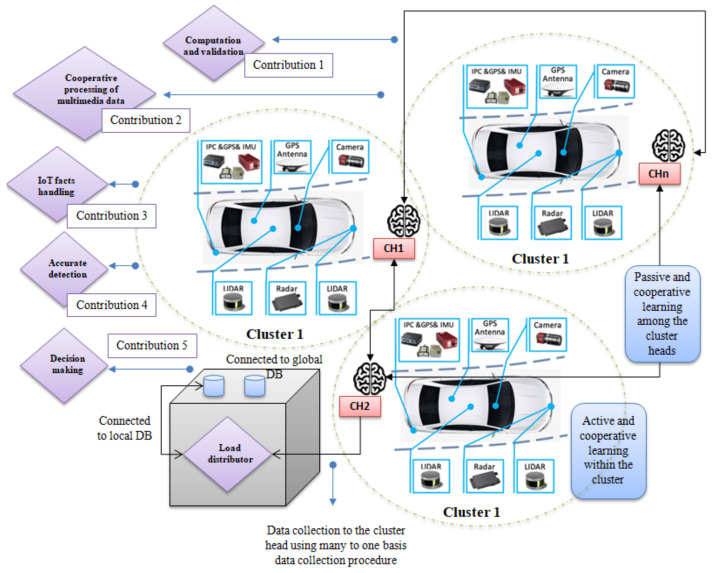
Proposed cooperative learning model for task slicing and load distribution among edge devices.

**Figure 2 sensors-22-04133-f002:**
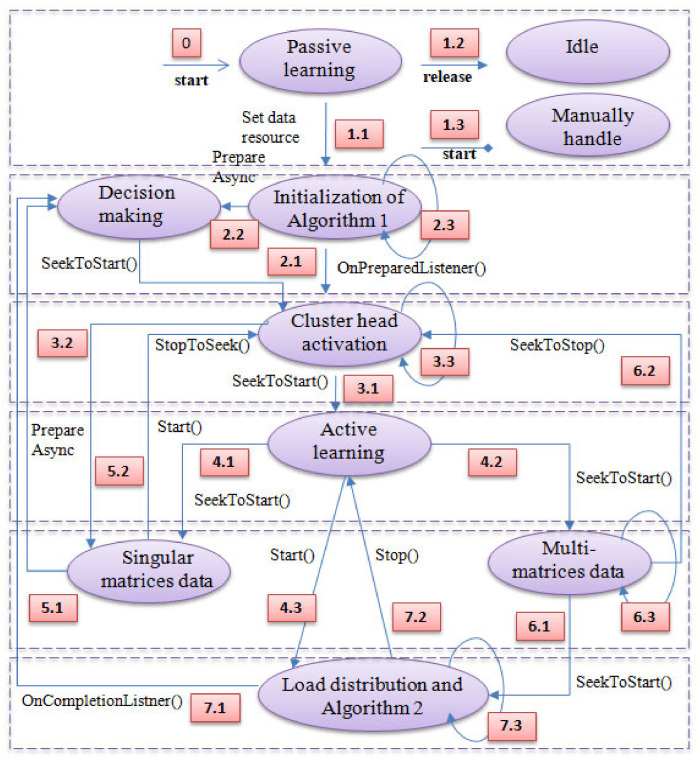
Conditional decision-making flow diagram for proposed edge intelligence-based cooperative learning model for industrial Internet of Things.

**Figure 3 sensors-22-04133-f003:**
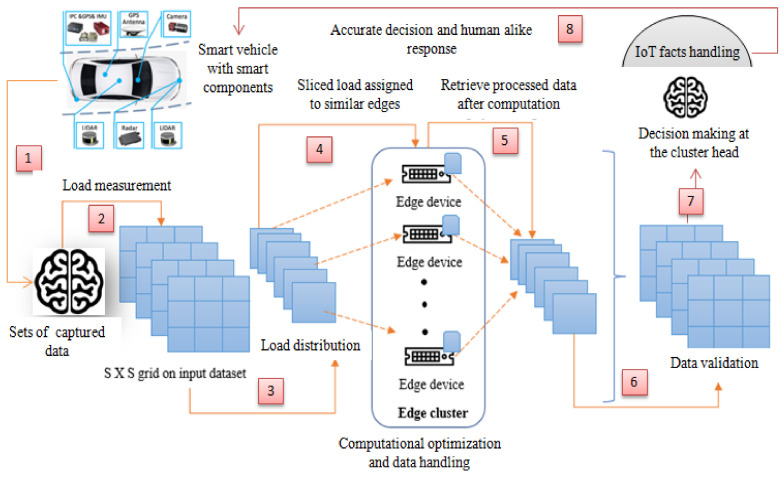
Work-flow diagram of proposed edge intelligence-based cooperative learning model with multimedia object-processing handler in industrial Internet of Things. The Figure 3 explains the flow diagram of the proposed work.

**Figure 4 sensors-22-04133-f004:**
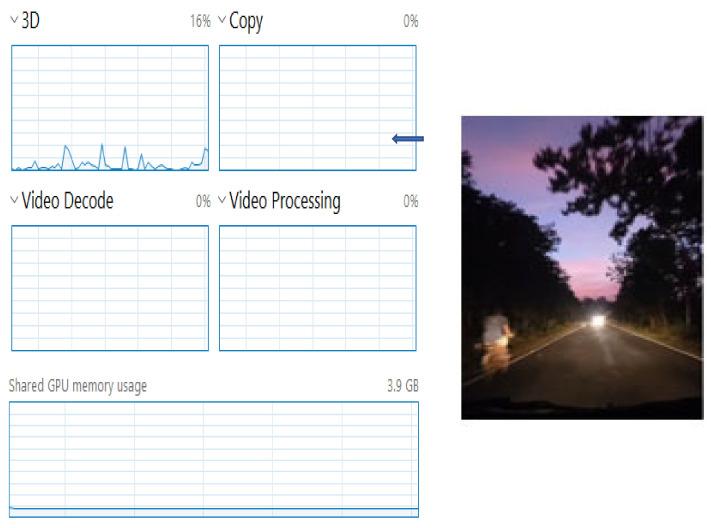
Utilization of processing unit at the initialization of Algorithm 1, decision making by the cluster head.

**Figure 5 sensors-22-04133-f005:**
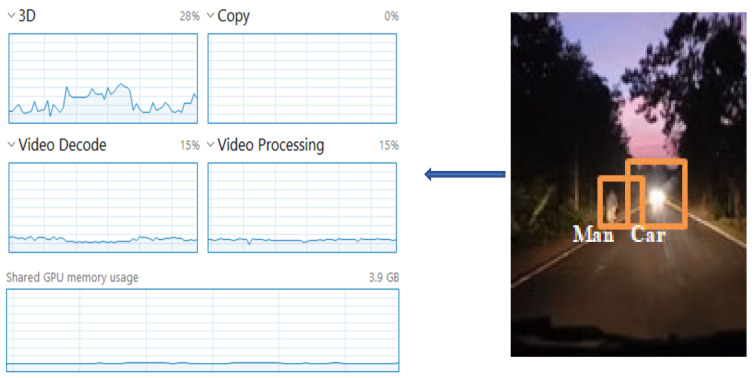
Utilization of processing after the initialization of image processing by the cooperative-based learning approach.

**Figure 6 sensors-22-04133-f006:**
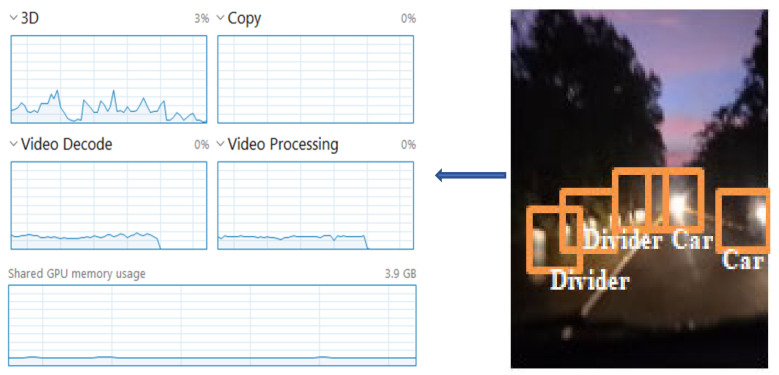
Utilization of processing after the initialization of video processing by the cooperative-based learning approach.

**Figure 7 sensors-22-04133-f007:**
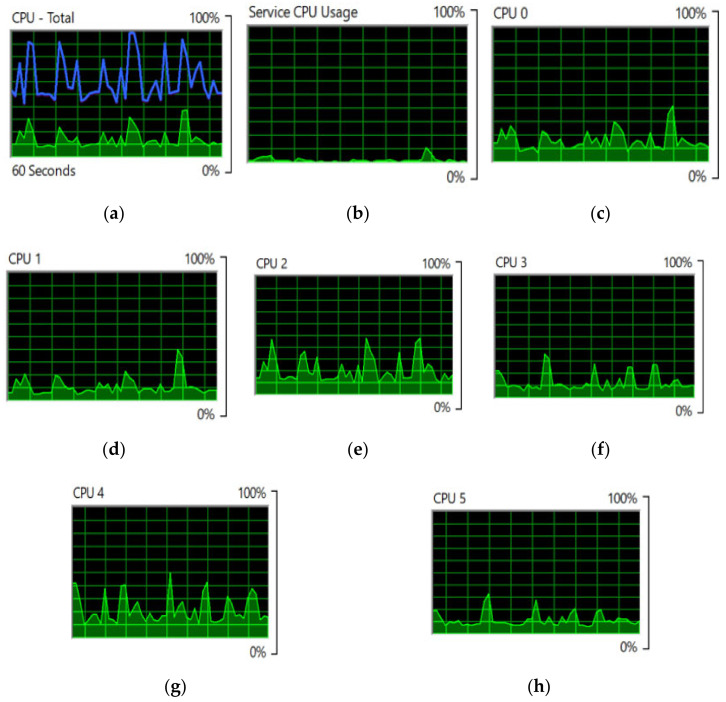
(**a**) Overall processor utilization for video-data processing by the cooperative-based learning approach. (**b**) Service-processor utilization of the mother edge. (**c**) Utilization of the first edge device for video-data processing by the cooperative-based learning approach. (**d**) Utilization of 2nd edge device for video-data processing by the cooperative-based learning approach. (**e**) Utilization of 3rd edge device for video-data processing. (**f**) Utilization of 4th edge device for video-data processing. (**g**) Utilization of 5th edge device for video-data processing. (**h**) Utilization of 3rd edge device for video-data processing.

**Figure 8 sensors-22-04133-f008:**
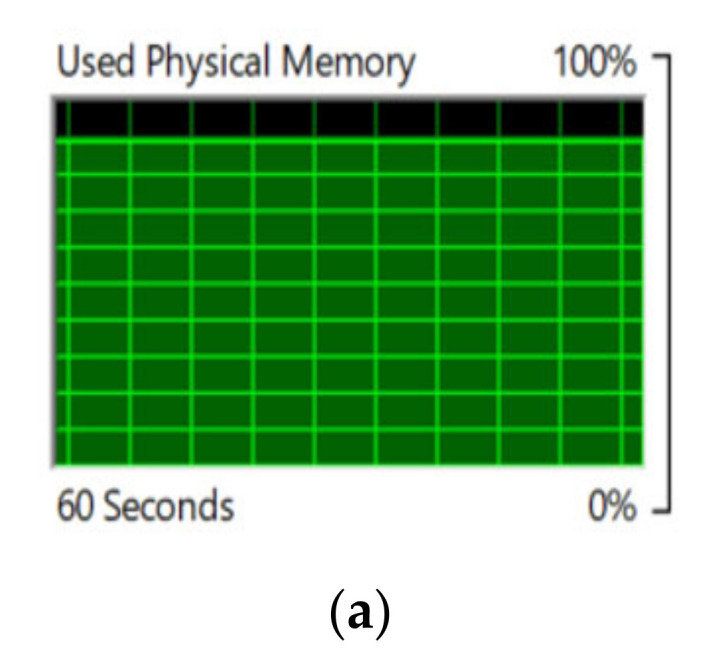
(**a**) Utilization of cluster-head memory while making a decision for near-optimized video data processing by the cooperative-based learning approach. (**b**) Utilization of virtual memory at cluster head while making a decision for near-optimized video data processing by the cooperative-based learning approach. (**c**) Hard faults/sec measurement by the cluster head while distributing the load into the edge cluster for near-optimized video-data processing by the cooperative-based learning approach.

**Figure 9 sensors-22-04133-f009:**
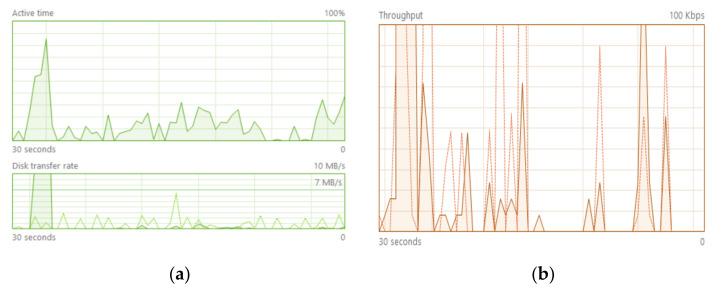
(**a**) Data-allocation rate from receiver end to the processing edge for near-optimized video-data processing by the cooperative-based learning approach. (**b**) Network throughput by the edge cluster.

**Figure 10 sensors-22-04133-f010:**
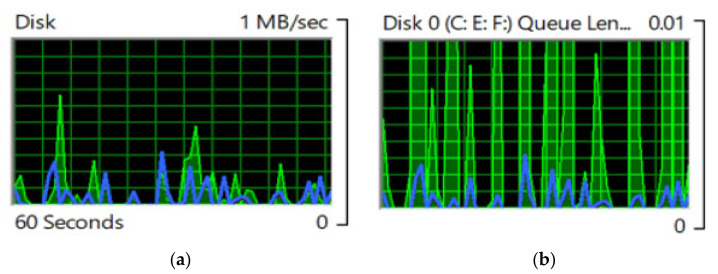
(**a**) Disk-utilization rate while transferring data from the surrogate end to the mother edge in the presence of the cluster head for near-optimized video-data processing by the cooperative-based learning approach. (**b**) Processed-data collection from surrogate edges to the mother edge in the presence of the cluster head for near-optimized video-data processing by the cooperative-based learning approach.

**Figure 11 sensors-22-04133-f011:**
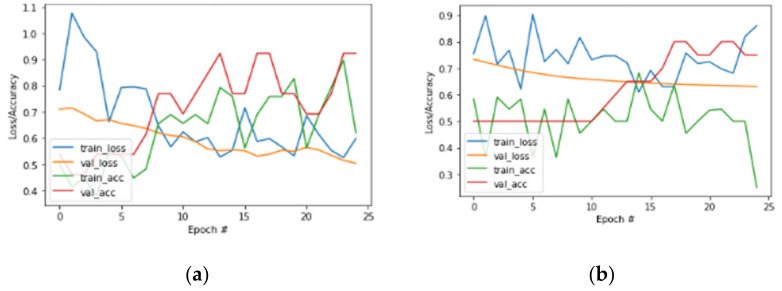
(**a**) Comparative loss/accuracy curve for existing DL models. (**b**) Comparative loss/accuracy curve for existing DL models.

**Figure 12 sensors-22-04133-f012:**
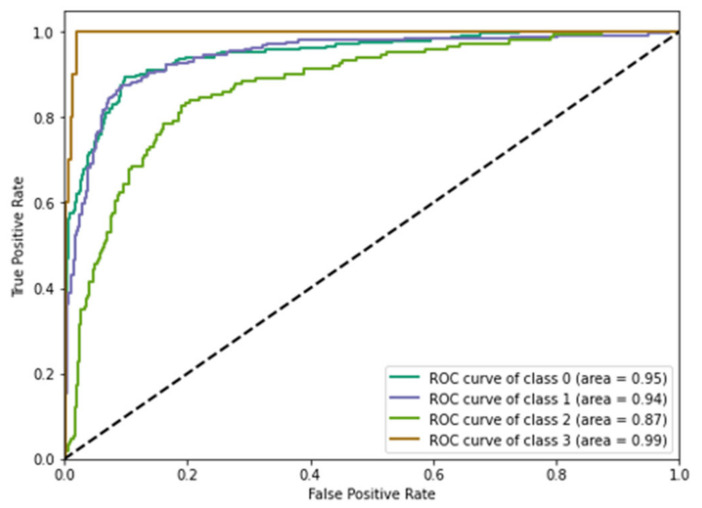
Receiver operating characteristic curve (ROC) for existing DL models.

**Table 1 sensors-22-04133-t001:** Summary of existing literature with their contributions and areas of application.

Ref. Number	Year of Publishing	Area of Application	Use	Remark
[1]	2020	Federated learning and edge computing	Survey of the probable application of federated learning	On macro perception
[2]	2019	Integrated Wireless Network (IWN)	Deep learning enables algorithms and specific security for analyzing traffic systems	On micro perception
[6]	2019	Edge intelligence	Deep learning-based application to train and test, improving edge interface system	On macro perception
[8]	2020	Edge intelligence	Improving edge interface by training, testing, and uploading	On macro perception
[14]	2021	Edge computing and artificial intelligence	Survey deep learning method and those applied to improve edge intelligence method	On macro perception
[16]	2019	Edge intelligence	Enhancing the cloud service using deep learning applications	On macro perception

**Table 2 sensors-22-04133-t002:** List of the components.

Sl. No.	Module Name	Compont Description	Number of Component
1	Camera module	Maximum image transfer rate 1080p: 30 fps (encode and decode) 720p: 60 fps, resolution 8-megapixels Still-picture resolution: 3280 × 2464	4
2	Sound-capture module	Sound card, comes with inbuilt microphone, compatible with the Raspberry Pi 4	4
3	IMU-enabled GPS device	Dimensions: 115 × 93 × 35 mm, 115 × 93 × 35 mm; weight, 330 g, 740 g; with accelerometer; gyroscope	1
4	Radar	2 (before and after) Wavelength: 3.2 cm Transmit frequency: 9.3–9.8 GHz Beamwidth (circular): 1.8°	1
5	Control unit	Raspberry Pi 4	4
6	User interface	Web interface using Django backend	1

**Table 3 sensors-22-04133-t003:** Literature summary of different prototypical compressions.

Ref. Number	Model Applied	Interface or Approach	Solution	Comparative Performance of Our Proposed Edge Intelligence-Based Cooperative Learning Model
[4]	ResNet	Compact-level modeling	Improve acceleration by training	Relatively improved
[8]	Faster CNN	Parameter tanning	Minimized mode size	Smaller than earlier
[9]	CNN	Information concentration and regularization	Minimized the storage utilization	More accurate
[11]	NN	Information distillation	Compress the model	Slightly improved
[12]	Google Net	Information distillation	Minimized the acceleration, and less storage utilization	Faster and advanced
[13]	DNN	E2C	Level segregation for better uploading	Improved
[14]	DNN and CNN	Kernal segregation	Minimized the acceleration, and less storage utilization	Faster and advanced

## Data Availability

Not applicable.

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
