# Peer review of "Improved Multimedia Object Processing for the Internet of Vehicles"

_sensors, 2022, doi:10.3390/s22114133_

Round 1

Reviewer 1 Report

Summary of the contributions:

  1. Authors recommend a more accurate and quicker object detection solution using intelligent cameras and other sensor ends of a smart car. The paper's objective is practical, and the solution could be adopted in Industry 4.0. 
  2. The proposed work involves edge cluster computing to make the IoV network more skilled in real-time object detection. Furthermore, edge intelligence has been incorporated in the case of task distribution and resultant task integration.
  3. The test bed evaluation and result analysis sections are also satisfactory to claim the solution objectives.

 The paper is technically sound and needs a few minor modifications.

Comments to the authors:

  • The organization and flow of the paper are acceptable. However, the authors are requested to review the paper again to improve grammatical potency. 
  • The problem description and the effect of the work are sufficiently verified. However, the adopted methodology could be more emphasized.
  • The caption of Table 2 should be revised.
  • Overall, the conception and contribution of the paper are appropriate for this journal. 

Author Response

Comment 1: Authors recommend a more accurate and quicker object detection solution using intelligent cameras and other sensor ends of a smart car. The paper's objective is practical, and the solution could be adopted in Industry 4.0. 
The proposed work involves edge cluster computing to make the IoV network more skilled in real-time object detection. Furthermore, edge intelligence has been incorporated in the case of task distribution and resultant task integration.
The test bed evaluation and result analysis sections are also satisfactory to claim the solution objectives.
The paper is technically sound and needs a few minor modifications.

Response: Thank you very much for your valuable suggestions. We have incorporated the given suggestions. We hope accept the revised version of the manuscript will be accepted.   

Comment 2: The organization and flow of the paper are acceptable. However, the authors are requested to review the paper again to improve grammatical potency. 

Response: Thank you very much for your valuable advices. We have thoroughly gone through the paper and tried to improve grammatical potency.

Comment 3: The problem description and the effect of the work are sufficiently verified. However, the adopted methodology could be more emphasized.

Response: Thank you for your suggestion. We have tried to establish the purpose of our proposal concerning the existing work through this particular section, highlighted with green color in the revised manuscript. We hope you will recognize our effort. 

Comment 4: The caption of Table 2 should be revised.

Response: Thank you for your valuable advice. As per your suggestion, the caption of Table 2 have been revised. 
Comment 5: Overall, the conception and contribution of the paper are appropriate for this journal.

Response: Thank you for your valuable consideration.

Reviewer 2 Report

-This paper is not well-written. The authors should revise the English writing. The flow between each sentence, paragraph and section, should be improved.

-I suggest to revise the title. It is too long. Please, clearly mention your main contribution at the beginning of the title.

-The authors did not define the full name of IoV and IoT in the abstract.

-All the abbreviations must be defined in their first appearance in the abstract and the main text.

-In Figure 1, you assign a number to each contribution. However, in the text, it is not clear which contribution is assigned.

-There is no discussion in the introduction to show that how your contributions can improve to the state of the art?

-At the end of introduction section, the authors should provide a summary for the rest of this paper. What do you want to do in each section?

-There are two sub-sections with 2.5 number.

-Between Section 2 and sub-section 2.1, explain that why the literature review is divided into different sub-sections? Add more justifications and clarifications.

-The literature review is very limited and many relevant works are ignored. I suggest to search recent published papers from popular scholars in the filed like Prof. Ugo Fiore, Dr. Fathollahi-Fard, Prof. Dulebenets and so on. For example, the following relevant works can be cited:

Blockchain in supply chain management: a review, bibliometric, and network analysis. Environmental Science and Pollution Research 

An integrated approach for a sustainable supplier selection based on Industry 4.0 concept. Environmental Science and Pollution Research,

-In Section 3, in Table 2, the authors defined the abbreviations. However, the list of abbreviations normally put in the appendix.

-There are more than one table with number 2.

-There are many figures which make the paper difficult to understand. There is not enough justification and explanation for each figure. It should be noted the quality of some charts is too low.

-In the conclusion section, we normally use the past verbs for the sentences as we did something in the paper. The authors must talk about their findings, limitations and recommendations.

-There is no future research recommendation in the conclusion section.

Author Response

Comment 1: This paper is not well-written. The authors should revise the English writing. The flow between each sentence, paragraph and section, should be improved.

Response: Thank you very much for your valuable suggestions. We have thoroughly gone through the paper and removed the typos and gave effort to make the paper understandable. Other suggestions also have been incorporated. We hope you will accept the revised version of the manuscript.   

Comment 2: I suggest to revise the title. It is too long. Please, clearly mention your main contribution at the beginning of the title.

Response: Thank you very much for your valuable suggestion. We have revised the paper title. We hope you will accept the revised title of the manuscript.   

Comment 3: The authors did not define the full name of IoV and IoT in the abstract.

Response: Thank you very much for your observation. We have defined the full name of IoV and IoT in the abstract of the revised manuscript version. 

Comment 4: All the abbreviations must be defined in their first appearance in the abstract and the main text.

Response: Thank you for your suggestion. We have done it accordingly. 

Comment 5: In Figure 1, you assign a number to each contribution. However, in the text, it is not clear which contribution is assigned.

Response: As per your recommendation, we have included assigned contributions, colored with blue in the revised version of the manuscript. 

Comment 6: There is no discussion in the introduction to show that how your contributions can improve to the state of the art?

Response: Thank you for your direction. The revised version of the paper includes the following into the introduction section. The added section is colored with blue color in the revised version of the manuscript.

Comment 7: At the end of introduction section, the authors should provide a summary for the rest of this paper. What do you want to do in each section?

Response: Thank you very much for your valuable suggestions. We have incorporated the paper organization at the end of the introduction, colour with blue, in the revised version of the manuscript.   

Comment 8: There are two sub-sections with 2.5 number.

Response: We have revised the numbers the subsections accordingly.   

Comment 9: Between Section 2 and sub-section 2.1, explain that why the literature review is divided into different sub-sections? Add more justifications and clarifications.

Response: Thank you very much for your valuable advices. We have organized the Introduction section as per your suggestion. We have summarized the contribution section, and the achievements are marked by yellow colour in the revised version of the paper. 

Comment 10: The literature review is very limited and many relevant works are ignored. I suggest to search recent published papers from popular scholars in the filed like Prof. Ugo Fiore, Dr. Fathollahi-Fard, Prof. Dulebenets and so on. For example, the following relevant works can be cited:
Blockchain in supply chain management: a review, bibliometric, and network analysis. Environmental Science and Pollution Research 
An integrated approach for a sustainable supplier selection based on Industry 4.0 concept. Environmental Science and Pollution Research.

Response: Thank you for your suggestion. As per your suggestion, we have included these recent and relevant papers with their discussion in the revised version of the manuscript, as our future work. 

Comment 11: In Section 3, in Table 2, the authors defined the abbreviations. However, the list of abbreviations normally put in the appendix.

Response: We support your valuable advice. But, our defined algorithms consist of these abbreviations forms in section 3. Therefore, we included the list of abbreviations in the same section to understand their complete forms. We hope the reason is justified.  

Comment 12: There are more than one table with number 2.

Response: We have revised the table numbers accordingly.   

Comment 13: There are many figures which make the paper difficult to understand. There is not enough justification and explanation for each figure. It should be noted the quality of some charts is too low.

Response: Thank you very much for your valuable advices. We have done the corrections accordingly. 

Comment 14: In the conclusion section, we normally use the past verbs for the sentences as we did something in the paper. The authors must talk about their findings, limitations and recommendations.

Response: Thank you for your suggestion. We have tried to establish the purpose, findings, limitations and recommendations of our proposal concerning the existing work through this particular sections. We hope you will recognize our effort. 

Comment 15: -There is no future research recommendation in the conclusion section.

Response: Thank you for your valuable advice. As per your suggestion, we have included future aspects of our research findings in the revised version of the manuscript. The additional portion has been colored in blue.

Reviewer 3 Report

This paper presents an interesting research topic for IoV. There are some observations.

  1. The paper comes to the proposed method in the introduction without developing the story and the importance of edge intelligence. The introduction is very short, and it is hard for the readers to find the motivation for this work.
  2. The same goes for the related work section. They are claiming the uniqueness of this work without referring to other works. Line 90-102.
  3. Only 16 references do not make enough to write a journal on such an important topic. Authors need to do more research on this topic. Recently there have been huge advancements in edge sensors.
  4. Line 98, what is the background to adapting slicing and stitching methodology? Can you provide a literature study prior to that?
  5. I can not fully agree with line 91. Yes, there are some problems. But the recent studies makes a sensor standalone with CNN. I can suggest some important papers with edge intelligence. 
    1. https://www.mdpi.com/1424-8220/21/5/1757/htm
    2. https://openaccess.thecvf.com/content_ICCV_2019/html/Bose_A_Camera_That_CNNs_Towards_Embedded_Neural_Networks_on_Pixel_ICCV_2019_paper.html
    3. https://ieeexplore.ieee.org/abstract/document/9443667        These three papers discuss how to bring intelligence camera. They deployed hardware-accelerated custom ASIC, FPGA to accelerate the overall processing. They also worked with a low-power design strategy by bio-inspired event-based processing. You should consider citing them and distinguish them from your work.
  6. The novelty lies on your 6 levels. You shoild provide some detailed description here. line 270, 273: what are other stitching?
  7. Edge is power sensative. Do you have any power saving estimations?
  8. Hardware description for edge device is missing. What kind of hardware platform was used for your experiment? A detailed description is required.
  9. Figures in a box does not look good.
  10. Your read speed is 472kb/s. What is the image size you used in the experiment? When the image size will increase, then can you be able to match the frame rate? There are lot of research performs event based processing. Please refer/cite event based processing to improve your proposed work.

Author Response

Comment 1: This paper presents an interesting research topic for IoV. There are some observations.

The paper comes to the proposed method in the introduction without developing the story and the importance of edge intelligence. The introduction is very short, and it is hard for the readers to find the motivation for this work.
The same goes for the related work section. They are claiming the uniqueness of this work without referring to other works. Line 90-102.

Response: Thank you very much for your valuable observations. We have tried to incorporate your suggestions. We hope you will accept the revised version of the manuscript.   

Comment 2: Only 16 references do not make enough to write a journal on such an important topic. Authors need to do more research on this topic. Recently there have been huge advancements in edge sensors. 

Response: Thank you very much for your valuable advices. We have gone through recent publications and incorporated those as per your suggestion. 

Comment 3: Line 98, what is the background to adapting slicing and stitching methodology? Can you provide a literature study prior to that?

Response: Thank you for your suggestion. We have tried to establish the purpose of our proposal concerning the existing work through this particular section. We hope you will recognize our effort. 

Comment 4: I can not fully agree with line 91. Yes, there are some problems. But the recent studies makes a sensor standalone with CNN. I can suggest some important papers with edge intelligence. 

https://www.mdpi.com/1424-8220/21/5/1757/htm
https://openaccess.thecvf.com/content_ICCV_2019/html/Bose_A_Camera_That_CNNs_Towards_Embedded_Neural_Networks_on_Pixel_ICCV_2019_paper.html
https://ieeexplore.ieee.org/abstract/document/9443667

These three papers discuss how to bring intelligence camera. They deployed hardware-accelerated custom ASIC, FPGA to accelerate the overall processing. They also worked with a low-power design strategy by bio-inspired event-based processing. You should consider citing them and distinguish them from your work.

Response: Thank you for your valuable advice. As per your suggestion, we have included these recent and relevant papers with their discussion in the revised version of the manuscript. 

Comment 5: The novelty lies on your 6 levels. You should provide some detailed description here. Line 270, 273: what are other stitching?

Response: Thank you for your direction. The revised version of the paper includes a detail about the levels, colored in yellow.

Comment 6: Hardware description for edge device is missing. What kind of hardware platform was used for your experiment? A detailed description is required.

Response: Thank you very much for your valuable suggestions. We have incorporated the descriptions of fundamental hardware platform. Detailed description in Table 6, is colored with yellow in the revised version of the manuscript.   

Comment 7: Figures in a box does not look good. 

Response: As per your suggestion we removed boundaries of all the boxes. 

Comment 8: Your read speed is 472kb/s. What is the image size you used in the experiment? When the image size will increase, then can you be able to match the frame rate? There are lot of research performs event based processing. Please refer/cite event based processing to improve your proposed work.

Response: Thank you for your queries. We took input camera module with the specification of max image transfer rate 1080p: 30fps (encode and decode), 720p: 60fps, resolution 8-megapixel, still picture resolution 3280 x 2464 and the read speed by the controller is a summation of audio visual data concerning the processing unit. We have noted your suggestions regarding the research performs toward event based processing and will include as our future work list, to improve our proposed work. 

Round 2

Reviewer 2 Report

Thank you so much for revising the paper. It is highly improved as compared to the initial draft. 

In the conclusion, one technical comment is to suggest the development of novel metaheuristics like red deer algorithm, whale optimization algorithm and social engineering optimizer for your research area. You can cite to their main papers. 

One minor comment is that we cannot find the number of references at the end of paper. Please, revise the format of the paper carefully to remove all the shortcomings. 

Thank you for your time and efforts to improve your paper and meet the standards of a high-ranked journal like Sensors. 

Author Response

Comment 1: Thank you so much for revising the paper. It is highly improved as compared to the initial draft.

In the conclusion, one technical comment is to suggest the development of novel metaheuristics like red deer algorithm, whale optimization algorithm and social engineering optimizer for your research area. You can cite to their main papers.

Response: Thank you very much for your valuable consideration. We have gone through the research works and incorporated them accordingly as per your suggestion. Changes made in the revised version have been colored in blue. We hope the revised version of the manuscript will be accepted.   

Comment 2: One minor comment is that we cannot find the number of references at the end of paper. Please, revise the format of the paper carefully to remove all the shortcomings.

Response: Thank you very much for your valuable suggestion. We beg apologies for such kind of inaccuracy. We have revised the manuscript thoroughly to overcome such shortcomings.   

Comment 3: Thank you for your time and efforts to improve your paper and meet the standards of a high-ranked journal like Sensors.

Response: Thank you very much for your kind consideration.

Reviewer 3 Report

Thank you authors for taking the time to improve the manuscript. The revised version meets most of the expectations. It needs some work to finalize.

  • Figures are still in the box and the representation is not okay. My suggestion is you can use only one figure number Figure 10 for example instead of Figure 10,11 and 12. Using a,b, and c you can separate them. The description will be Figure 10 (a) Utilization of 2nd edge device for video data processing by cooperative-based learning approach (b)...(c)... This instruction is applicable to all figures. 
  • Reference numbers are dotted. So I can't correlate the reference and text. Please fix them.

Author Response

Comment 1: Thank you authors for taking the time to improve the manuscript. The revised version meets most of the expectations. It needs some work to finalize.

Figures are still in the box and the representation is not okay. My suggestion is you can use only one figure number Figure 10 for example instead of Figure 10,11 and 12. Using a,b, and c you can separate them. The description will be Figure 10 (a) Utilization of 2nd edge device for video data processing by cooperative-based learning approach (b)...(c)... This instruction is applicable to all figures.

Response: Thank you very much for your valuable observations and consideration. We have tried to incorporate your suggestions. Changes made in the revised version has been colored in yellow. We hope you will accept the revised version of the manuscript.   

Comment 2: Reference numbers are dotted. So I can't correlate the reference and text. Please fix them.

Response: Thank you very much for your valuable suggestion. We beg apologies for such kind of inaccuracy. We have revised the manuscript thoroughly to overcome such shortcomings.